# Effect of LDHA Inhibition on TNF-α-Induced Cell Migration in Esophageal Cancers

**DOI:** 10.3390/ijms232416062

**Published:** 2022-12-16

**Authors:** Agata Forkasiewicz, Wojciech Stach, Jaroslaw Wierzbicki, Kamilla Stach, Renata Tabola, Anita Hryniewicz-Jankowska, Katarzyna Augoff

**Affiliations:** 1Department of Surgical Education, Wroclaw Medical University, 50-368 Wroclaw, Poland; 2Department of Biochemistry and Immunochemistry, Wroclaw Medical University, 50-368 Wroclaw, Poland; 3Department of Minimally Invasive Surgery and Proctology, Wroclaw Medical University, 50-556 Wroclaw, Poland; 4Department of Thoracic Surgery, Wroclaw Medical University, 53-439 Wroclaw, Poland; 5Laboratory of Cytobiochemistry, Biotechnology Faculty, University of Wroclaw, 50-383 Wroclaw, Poland

**Keywords:** lactate dehydrogenase, sodium oxamate, TNF-α–ERK1/2 signaling pathway, metalloproteinase 9, cancer cell migration, esophageal cancer

## Abstract

Cell migration is an essential part of the complex and multistep process that is the development of cancer, a disease that is the second most common cause of death in humans. An important factor promoting the migration of cancer cells is TNF-α, a pro-inflammatory cytokine that, among its many biological functions, also plays a major role in mediating the expression of MMP9, one of the key regulators of cancer cell migration. It is also known that TNF-α is able to induce the Warburg effect in some cells by increasing glucose uptake and enhancing the expression and activity of lactate dehydrogenase subunit A (LDHA). Therefore, the aim of the present study was to investigate the interrelationship between the TNF-α-induced promigratory activity of cancer cells and their glucose metabolism status, using esophageal cancer cells as an example. By inhibiting LDHA activity with sodium oxamate (SO, also known as aminooxoacetic acid sodium salt or oxamic acid sodium salt) or siRNA-mediated gene silencing, we found using wound healing assay and gelatin zymography that LDHA downregulation impairs TNF-α-dependent tumor cell migration and significantly reduces TNF-α-induced MMP9 expression. These effects were associated with disturbances in the activation of the ERK1/2 signaling pathway, as we observed by Western blotting. We also reveal that in esophageal cancer cells, SO effectively reduces the production of lactic acid, which, as we have shown, synergizes the stimulating effect of TNF-α on MMP9 expression. In conclusion, our findings identified LDHA as a regulator of TNF-α-induced cell migration in esophageal cancer cells by the ERK1/2 signaling pathway, suggesting that LDHA inhibitors that limit the migration of cancer cells caused by the inflammatory process may be considered as an adjunct to standard therapy in esophageal cancer patients.

## 1. Introduction

Esophageal cancer (EC), represented by two major subtypes, esophageal squamous cell carcinoma (ESCC) and esophageal adenocarcinoma (EAC), is one of the most aggressive, rapidly metastatic cancers, ranking sixth in terms of malignant mortality and eighth in terms of incidence worldwide [1]. However, there are marked differences in the geographic distribution of the two subtypes around the world. The highest incidence of ESCC is found in sub-Saharan Africa (e.g., Malawi, Zimbabwe, and Uganda), Central Asia (e.g., Mongolia, Turkmenistan, and Azerbaijan), and East Asia, with China leading in terms of incidence at the national level, whereas EAC is most common in developed countries, with the highest incidence in North America, Northern and Western Europe, and Oceania [2,3]. In contrast to EAC, where numbers are steadily increasing, ESCC accounts for 90% of all diagnosed esophageal cancers, although there has been a slight decline in new cases over the past 40 years [4,5].

Major risk factors for esophageal cancers include smoking, an unhealthy diet low in micronutrients, obesity, consumption of alcohol and hot liquids, and Barrett’s esophagus and gastroesophageal reflux disease, associated mainly with EAC [6]. Most of these factors are the cause of inflammation, and many of them can lead to chronic inflammation. It is now widely accepted that inflammatory responses, regulated by various pro-inflammatory cytokines, including tumor necrotic factor-α (TNF-α), play an important role in cancer development and progression. TNF-α, a prototypical pro-inflammatory cytokine with a broad spectrum of biological functions, was initially recognized only as an inhibitor of tumor cell proliferation and inducer of apoptosis. However, over time it has been shown that TNF-α may also have a stimulatory effect on the survival and motility of many different types of cancer cells, including EC cells, and act as an endogenous promoter of carcinogenesis, which is probably due to the complex role of two TNF-α specific receptors, tumor necrosis factor receptor 1 (TNFR1), also known as tumor necrosis factor receptor superfamily member 1A (TNFRSF1A) and CD120a, and tumor necrosis factor receptor 2 (TNFR2), also known as tumor necrosis factor receptor superfamily member 1B (TNFRSF1B) and CD120b [7,8,9]. TNF-α was found to act as a major mediator of the expression of matrix metalloproteinase 9 (MMP9), one of two gelatinases crucial for cell migration, tissue stroma organization, and new blood vessel formation [10,11]. MMP9, which degrades major extracellular structures and releases various signal peptides, is involved in tumor growth and metastasis by regulating epithelial-mesenchymal transition (EMT), stimulating tumor cell survival and invasion in response to TNF-α-induced inflammatory signals [12].

Recent studies have shown that TNF-α can regulate and alter cellular metabolism, mainly lipid metabolism [13]. In cancer cells, TNF-α was found to induce the Warburg effect, increasing aerobic glycolysis, ATP production, and lactate secretion and decreasing oxidative metabolism [14]. Previous observations have shown that during stem cell differentiation, TNF-α enhanced glucose uptake and increased the expression and activity of lactate dehydrogenase subunit A (LDHA), which correlated with increased lactate production [15]. Similarly, in non-Hodgkin’s lymphoma studies, it was found that TNF-α induced changes in the intracellular profile of LDH isoenzymes in lymphocytes and redirected pyruvate metabolism to lactate production, which was closely related to the degree of cell activation and response to chemotherapy [16]. LDH isoenzymes enriched in the LDHA subunit, such as LD5 and LD4, are thought to play an important role in maintaining glycolysis after bypassing the “Pasteur effect” in cancer cells [17,18]. Increased levels of LDHA are frequently observed in human malignancies, and silencing LDHA expression or using specific inhibitors against LDHA activity in some tumor cells was found to reduce lactate secretion, cell viability, and their ability to invade [19,20,21].

In light of this knowledge, we hypothesized that there is a direct link between the metabolic reprogramming of cells undergoing tumor transformation and their specific response to TNF-α, resulting in increased migratory potential and consequently increased invasiveness and metastatic capacity. Therefore, the purpose of this work was to address the question of whether inhibition of LDHA activity may affect the pro-migratory effects of TNF-α in esophageal cancers, with particular attention to the role of MMP9.

## 2. Results

### 2.1. LDHA Is Upregulated in Esophageal Cancer Tissues

To demonstrate the importance of LDH subunit A in esophageal tumor growth, we analyzed the expression level of LDHA genes in esophageal tumor tissue samples and compared it to the level determined for normal control tissues. Results indicated a significant increase in LDHA in tumors (mean = 14.04 ± 0.04) compared to normal tissues (mean = 13.62 ± 0.07; *p* < 0.005) (Figure 1A). Then, to trace the distribution of LDHA in esophageal normal (n = 6) and cancer tissues (n = 9), paraffin sections were stained by IHC. Normal epithelial cells stain very weakly or not at all, in contrast to cancer cells derived from both squamous and glandular epithelium (Figure 1B). Tumor cells generally stain intensely for LDHA, although unevenly within the same cluster of tumor cells.

### 2.2. TNF-α Induces an Increase in LDHA and LDHB Gene Expression

To assess whether TNF-α is involved in the regulation of LDH in esophageal cancer cells, we analyzed the expression levels of genes encoding LDH subunits (LDHA and LDHB) in TNF-α-treated KYSE150 and EC7 cells versus untreated cells using qPCR. An increase in LDHA and LDHB gene expression was observed in TNF-α-treated KYSE150, where the FC value was 1.98 ± 0.08 and 1.88 ± 0.2; *p* < 0.005, for LDHA and LDHB, respectively. EC7 cells, such as KYSE150, significantly increased LDHA expression when exposed to TNF-α (FC value was 2.28 ± 0.26; *p* < 0.001). However, for LDHB, the TNF-α-induced increase in expression levels was small, with an FC of 1.47 ± 0.39, and not statistically significant compared to untreated control cells (Figure 2A). Basal expression levels of LDHA and LDHB in KYSE150 and EC7 cells were determined by regular PCR and are shown in Figure 2B.

In KYSE150 cells, the TNF-α-stimulated increase in expression of both LDH subunits was at a similar level and did not alter the percentage composition of LDH isoforms, as confirmed by zymography (Figure 3A). The calculated average value of the A/B ratio in KYSE150 cells was 2.11 ± 0.09 and did not change under the influence of TNF-α. In EC7 cells, the basal level of LDHB expression was found to be very low, and only the LD5 isoform was observed after visualization of zymography gels in both TNF-α-treated and untreated cells. (Figure 3B). Densitometric analysis of these gels showed a dose-dependent slight effect of TNF-α on the increase in cellular LD5 activity. TNF-α-induced changes in LDHA and LDHB protein levels in EC7 and KYSE150 cells are shown in Figure 3C. As we observed, TNF-α increased not only the level of LDHA and LDHB proteins but also tyrosine residue (Tyr10) phosphorylation, which correlated with increased HIF-1α levels in cells. These changes also correlated with an increase in MMP9 expression. In addition, in the presence of TNF-α, EC7 cells showed a tendency to increase lactic acid secretion (FC = 1.26 ± 0.15).

To confirm that SO does not affect the LDH isoenzyme patterns, we treated KYSE150 cells with oxamate at different concentrations, 25 mM and 50 mM. As shown in Figure 3D, SO did not change isoenzymatic composition, also in the presence of TNF-α.

### 2.3. Sodium Oxamate (SO) Decreases Secretion of Lactic Acid and Inhibits TNF-α-Induced Cell Migration

The addition of SO at concentrations of 25 and 50 mM to the culture medium did not affect the viability of either KYSE150 or EC7 cells within 48 h, regardless of whether they were additionally treated with TNF-α or not, as shown by the MTT assay (Figure 4A). In contrast, SO significantly reduced lactate production in esophageal cancer cells, even when the cells were additionally stimulated with TNF-α. SO lowered the level of LA released by KYSE150 from 16.3 ± 0.9 µM to 5.5 ± 0.9 µM and from 16.96 ± 1.5 µM to 6.96 ± 1.5 µM when the cells were exposed to TNF-α. A similar effect was observed in EC7 cells, where SO reduced LA secretion from 10.5 ± 0.7 µM to 2.4 ± 0.7 µM in the absence of TNF-α and from 12.7 ± 2.2 µM to 2.9 ± 0.85 µM in the presence of TNF-α (Figure 4B). However, in contrast to SO, siRNA-mediated silencing of the LDHA gene in the cells alone resulted in only a slight reduction in lactic acid release, and the observed differences were not statistically significant (Figure 4C).

Analyzing the effect of SO on TNF-α-induced cell migration, it was found that the SO effectively abolished the pro-migratory activity of TNF-α on KYSE150 cells (Figure 5A). In the case of control cells, the ratio of wound closure was 0.3 ± 0.05 after 24 h and 0.7 ± 0.05 after 48 h, and 0.77 ± 0.06 and 0.99 ± 0.0004, respectively, after TNF-α stimulation. In the presence of SO, the wound closure rate was only 0.006 ± 0.005 after 24 h and 0.08 ± 0.02 after 48 h, and after TNF-α stimulation, 0.77 ± 0.06 and 0.99 ± 0.0004, respectively. After administration of TNF-α, the rate of wound closure in KYSE150 cells treated concurrently with SO was comparable to cells treated with SO alone and was 0.03 ± 0.02 and 0.05 ± 0.03 after 24 and 48 h, respectively.

To demonstrate a direct role for the LD5 isoform in the inhibitory effect of SO on TNF-α-stimulated cell migration, we used siRNA-mediated silencing of LDHA gene expression. Although siRNA-mediated silencing of LDHA in KYSE150 cells (with silencing efficiencies up to 38 ± 1.5% of control levels) impaired TNF-α-induced cell migration, the observed differences from control cells were statistically insignificant (Figure 5B). After 24 h, the wound closure rate was 0.44 ± 0.05 in cells transfected with scramble sequence and 0.58 ± 0.01 in cells transfected with LDHA siRNA and 0.9 ± 0.03 and 0.7 ± 0.12, respectively, after TNF-α stimulation.

As shown by the zymograms, while SO did not disrupt the ratio in the composition of isoenzymes of LDH in cells, the use of LDHA siRNA shifted the percentage of isoforms in total LDH activity toward forms with an increased proportion of the B subunit, primarily LD1 (Figure 6A). This shift, however, was observed only in KYSE150. In EC7 cells, LDHA siRNA reduced LD5 isoform levels up to 55.7 ± 13%, but still only this form was visible in the zymograms (Figure 6B).

### 2.4. Oxamate Inhibits TNF-α-Induced MMP9 Expression by Affecting ERK1/2 Signal Transduction

Our previous studies have shown that TNF-α regulates the migration of cancer cells, including esophageal cancer, by affecting the expression of MMP9 [8,22]. In this study, we found by gelatin zymography that the treatment of both KYSE150 and EC7 with SO decreased TNF-α-induced MMP9 secretion in a dose-dependent manner (Figure 7A,B). By analyzing TNF-α-activated MMP9-related signaling pathways, we found that the level of TNF-α-induced ERK1/2 phosphorylation at Thr202/Tyr204 amino acid residues, but not NFκB/p65 at Ser536, was significantly reduced in SO-treated KYSE150 cells (Figure 7A), as well as in SO-treated EC7 cells (Figure 7B) compared to untreated cells.

An effect similar to that obtained after the application of SO was observed in cells with reduced LDHA expression. Cells transfected with siRNA targeting LDHA under TNF-α stimulation produced MMP9 at lower levels in comparison to untransfected control cells and cells transfected with random-sequence scrambled siRNA (Figure 8A,B). Densitometric analysis of changes in activation of ERK1/2 and NFκB signaling pathways showed that silencing of LDHA gene expression resulted in significantly impaired ERK1/2 phosphorylation in response to TNF-α (FC vs. cells untreated with TNF-α was 0.57 ± 0.04 for non-transfected cells; 0.45 ± 0.03 for LDHA siRNA transfected and 0.62 ± 0.04 for scrambled sequence siRNA transfected cells; *p* < 0.05) in KYSE150 cells (Figure 8A) and in EC7 cells (Figure 8B) where FC vs. cells untreated with TNF-α was 0.84 ± 0.04 for non-transfected cells; 0.71 ± 0.01 for LDHA siRNA transfected and 1.11 ± 0.03 for scrambled sequence siRNA transfected cells; *p* < 0.05. Such disturbances were not observed for TNF-α-induced NFκB/p65 activation.

### 2.5. Lactate Acid Enhances TNF-α-Induced Cell Migration and MMP9 Expression in Esophageal Cancer Cells

Previous reports suggest that high levels of exogenous lactate can affect cell motility, so in this work, we used a wound healing assay in which cells were pretreated with LA for 24 h to analyze the effect of lactic acid on the response of cancer cells to TNF-α [23]. As observed, the application of lactate at a concentration of 5 mM enhanced the promigratory effect of TNF-α in KYSE150 cells, and the differences with respect to the LA-untreated control were statistically significant (*p* < 0.05) (Figure 9A). In the presence of LA, the cells also increased MMP9 expression with the dose used (FC vs. LA untreated cells was 1.37 ± 0.19 and 1.58 ± 0.19 for LA at concentrations of 2.5 and 5 mM, respectively), as we discovered by gelatin zymography (Figure 9B).

## 3. Materials and Methods

### 3.1. Analysis of LDHA Gene Expression in Esophageal Cancer Tissues

Gene Expression database of Normal and Tumor tissues 2 (GENT2) (http://gent2.appex.kr) was used (19 January 2022) to extract data representing the expression levels of the LDHA gene, expressed by log2FC, in normal (n = 24) and esophageal cancer (n = 236) tissue samples, which were generated by the U133Plus2 microarray platform (GPL570) and made available in NCBI’s public database repository, GEO [24]. Comparative analysis was performed by Student’s t-test using STATISTICA software licensed by Wroclaw Medical University, Wroclaw, Poland (version 13.1; TIBCO Software Inc.). Significance was defined as *p* < 0.05.

### 3.2. Cell Culture and Treatment

The study was approved by the Wroclaw Medical University Ethics Committee (Title: Molecular mechanism of the regulation of TNF-α-induced MMP9 gene expression by CDKN1A and its role in invasiveness and metastasis in ESCC. KB number: 156/2018). The methods were carried out in accordance with the approved guidelines, and written informed consent was obtained prior to the study.

Human esophageal squamous cell carcinoma cell line, KYSE150, was purchased from Merck Leibniz-Institute DSMZ (ACC-No 375). The EC7 cell line was established from an esophageal biopsy specimen taken at the time of esophageal prosthesis from a 68-year-old Caucasian male with recurrent esophageal adenocarcinoma. The patient was previously treated with systemic therapy and radiation therapy and died twenty-eight months after first diagnosis of metastatic disease.

The tissue sample taken from ESC p”tien’ was delivered to the laboratory in MEM medium within 2 h of collection, transferred to 3 cm plates, and minced with a scalpel into smaller pieces (sizes at most 1 mm^3^). The entire content has been transferred to a 15 mL tube and centrifuged at 900× *g* for 5 min. The pellet was washed twice with 2.5 mL of RPMI-1640/ Ham’s F12 medium (1:1) containing 2 mM L-glutamine, 20% FBS, antibiotics (200 U/mL penicillin, 10 µg/mL gentamicin) and amphotericin B at high concentration (10 µg/mL). It was then placed on a 3 cm cell culture dish in RPMI-1640/ Ham’s F12 medium (1:1) containing 2 mM L-glutamine and 20% FBS, 200 U/mL penicillin, 10 µg/mL gentamicin and 2.5 µg/mL amphotericin B. The culture was maintained at 37 °C in an atmosphere humidified with 5% carbon dioxide in air until adherent cells had attached to the plate surface and began to form colonies. Then the residual tissue structures were removed, and when the cells reached 80–90% confluence, passaging was started. Most fibroblasts were removed by differential trypsinization. After adaptation to culture conditions, the cells were maintained in RPMI-1640/ Ham’s F12 medium (1:1) with 2 mM L-glutamine and 10% FBS, without antibiotics. After four weeks, nine colonies of epithelial cells were separated from the culture using glass cylinders and maintained in continuous culture until the 45th passage, when it was assured that they were free of fibroblasts. One of the clones (C7), after determining the Short Tandem Repeat (STR) profile by ATCC Cell Line Authentication Service, was used for further experiments (Appendix A).

In experiments in which TNF-α alone or in combination with sodium oxamate (SO) (Santa Cruz, Dallas, TX, USA, sc-215880) or DL-lactic acid (LA) sodium salt solution (Fluka, Darmstadt, Germany, 71723) was used, cells were previously starved overnight in serum-free medium, and then treated with human recombinant TNF-α (Prospec, Ness-Ziona, Israel, cyt-223-b) at a concentration of 30 ng/mL for 24 h unless otherwise specified. The conditioned media (CM) were collected, suspended in 2x non-reducing sample buffer (62.5 mM Tris-HCl, pH 6.8, with 10% glycerol, 2% SDS, 0.05% bromophenol blue) and used immediately in gelatin zymography. Cell lysates were prepared using lysis buffer (25 mM NaH_2_PO_4_/Na_2_HPO_4_ buffer, pH 7.4 with 0.3 M NaCl and 1% triton X-100) supplemented with a cocktail of protease and phosphatase inhibitors (Cell Signaling, Danvers, MA, USA, 5872). Cell lysates were centrifuged at 16,000× *g* for 15 min at 4 °C. The supernatants were collected, and the protein concentration was determined using a Precision Red Advanced Protein Assay (Cytoskeleton, Denver, CO, USA, ADV02-A). For LDH zymography, samples were mixed 1:1 with 2x loading buffer (60% sucrose in HEPES buffer pH 8.0) or reduced with 4× Laemmli sample buffer (Bio-Rad, Warszawa, Poland, 161-0747) at 95 °C for 10 min for Western blotting and stored at −20 °C until use.

The effect of LD5 inhibition on cell viability in the presence and absence of TNF-α was determined using Trevigen’s TACS MTT Cell Proliferation Assay (R&D Systems, Minneapolis, MN, USA, 4890-050-K) according to the manufacturer’s instructions. Lactate concentration was determined in CM by a colorimetric assay performed in 0.2 M Tris–HCl buffer (pH 8.2) containing 1.3 mM β-NAD^+^, 0.66 mM nitrotetrazolium blue chloride (NBT), 0.28 mM phenazine methosulfate (PMS) and LDH enzyme (Sigma-Aldrich, Darmstadt, Germany, SAE0049-10KU) on 96-well plate and read at λ = 580 nm.

### 3.3. Zymography

In order to test MMP9 activity, CM samples were electrophoresed in 10% native polyacrylamide gel, containing 2 mg/mL gelatin. The gels were washed twice with 50 mM Tris-HCl (pH 7.5) buffer supplemented with 2.5% Triton X-100 and incubated overnight at 37 °C in 50 mM Tris-HCl (pH 7.5) buffer containing 150 mM NaCl, 10 mM CaCl_2_, 1 μM ZnCl_2_ and 0.05% Brij-35. Gels were stained with 0.125% (*w*/*v*) Coomassie Brilliant Blue R-250 in 62.5% (*v*/*v*) methanol and 25% (*v*/*v*) acetic acid solution, and de-stained in 10% acetic acid, 50% methanol. Gelatinolytic activity was visualized as unstained bands on a dark blue background.

In order to analyze the LDH isoenzyme composition, native cell extracts were electrophoresed in 7.5% discontinuous native polyacrylamide gel. In order to visualize the LDH isoenzyme bands, the gels were placed in a staining solution containing 0.5 mg/mL NAD^+^, 2.5 mg/mL nitrobluetrazolium (NBT), 2 mg/mL phenazinomethosulfate (PMS), 0.01 M sodium lactate and 0.03 mg/mL MgSO_4_ × 6H_2_O in 0.05 M Tris-HCl buffer (pH 7.5) and allowed to stain for 3–5 min in the dark at room temperature. Digital images were acquired using the Azure 280 detector. Pixel densities were analyzed with ImageJ software (in the period from 1 January 2022 to 30 November 2022) (https://imagej.nih.gov/ij/). LDH isozyme activity was calculated as a percentage of total LDH activity. The ratio of polypeptide A to B was calculated according to the method described by Stagg et al. using the formula: A = [(LDH5) + 0.75 (LDH4) + 0.50 (LDH3) + 0.25 (LDH2)/Σ(LDH1–5)] × 100, and B = 100 − A [25].

### 3.4. Immunohistochemistry (IHC)

Paraffin-embedded Section 4 and Section 5 μm thick were deparaffinized in two changes of xylene (10 min each), then rehydrated in decreasing concentrations of ethanol (95%, 70%, and 50%). For antigen recovery, slides were heated in 10 mM sodium citrate buffer pH 6.0 at 97 °C in a water bath for 40 min. Endogenous peroxidase was then blocked with 3% hydrogen peroxide in PBS for 10 min at room temperature. Tissue sections were immunostained using anti-LDHA antibody at a dilution of 1 µg/mL first and then Mouse/Rabbit IgG VisUCyte HRP Polymer Antibody (R&D Systems, VC002-025). LDHA-positive areas were visualized using 3,3′-diaminobenzidine. Mayer’s hematoxylin was used to counterstain the sections. Stained specimens were viewed under a light microscope (OlYMPUS CX43), and random areas per each section were captured as digital images (1296 × 972 pixels) with a digital camera (Moticam 5+, 5.0 MP) at 20x objective magnification (PlanC N 20×/0.4).

### 3.5. Western Blotting

For the Western blot analysis, the whole cell extracts were separated by SDS-PAGE and transferred to 0.2 μm nitrocellulose membranes (Bio-Rad, 1620112). Membranes were blocked with 5% nonfat milk in PBS, followed by overnight incubation at 4 °C with an appropriate primary antibody diluted in PBS.

Rabbit anti-β-Actin (4970), rabbit anti-HIF-1α (36169), mouse anti-ERK1/2 (4696), rabbit anti-Phospho-ERK1/2 (Thr202/Tyr204) (4370), mouse anti-NFκB p65 (6956), rabbit anti-Phospho-NFκB p65 (Ser536) (3033), rabbit anti-Phospho-LDHA (Tyr10) (8176) antibodies were products of Cell Signaling. Mouse anti-LDHA and rabbit anti-LDHB antibodies were acquired from Santa Cruz and Proteintech, respectively.

After three washes with 0.1% Tween-20 in PBS (PBS-T), membranes were incubated with the appropriate horseradish peroxidase-conjugated secondary antibody. Rabbit anti-mouse IgG (115-035-003) was a product of Jackson, and goat anti-rabbit IgG (7074) was obtained from Cell Signaling. Immunocomplexes were visualized by chemiluminescence in 100 mM Tris buffer pH 8.5 supplemented with p-coumaric acid (Sigma, 9008), luminol (Sigma, A891), and H_2_O_2_ using an Azure 280 detector. After stripping with mild stripping buffer (0.2 M glycine pH 2.2, 10% Tween 20, 1% SDS) the membranes were re-probed with other antibodies.

### 3.6. Quantitative Polymerase Chain Reaction (qPCR)

Total RNA was isolated from KYSE150 and EC7 cells using the RNeasy Mini Kit (Qiagen, 74104, Wroclaw, Poland) according to the manufacturer’s instructions. cDNA was synthesized from 2 μg of total RNA using the SuperScript IV VILO Master Mix with ezDNase Enzyme (Invitrogen, 11766050). PCR was performed with OneTaq^®^ Hot Start DNA Polymerase (New England BioLabs, M0481). Cycling conditions were as follows: initial denaturation at 94 °C for 30 s, 30 cycles of 30 s at 94 °C and 30 s at 60 °C, and final extension at 68 °C for 5 min. Each sample was run in triplicate. Amplification efficiency was assessed for all primer sets used in separate reactions, and primers with efficiencies in a range of 95–100% were used. Primers for the LDHA and LDHB genes were products of OriGene (HP208683 and HP208217). In order to normalize gene expression, primers for GAPDH or β-actin (ACTB) (OriGene, HP100003 and HP204660, respectively) were used. The ΔCt was calculated by subtracting the cycle threshold (Ct) value of GAPDH mRNA from the Ct value of the gene of interest (ΔCt). Fold change (FC) in the gene expression was calculated using the 2^−ΔΔCt^ method.

### 3.7. Wound Healing Assay

KYSE150 cells were plated on 12-well plates and grown to 70–75% confluence in complete medium, and then they were serum-starved overnight. The cell-free gaps were created by the ibidi Culture-Insert 3 Well (80369). Cells were then treated with 30 ng/mL of TNF-α or 50 mM SO and LA (5 mM) separately or after mixing in serum-free medium. KYSE150 cells with previously silenced expression of LDHA subunit were treated only with TNF-α. The pictures (2464 × 2056 pix) were captured at 12, 24, and 48 h at 20× magnification using a Zeiss digital camera integrated with an Axio Vert.A1 inverted microscope (Carl Zeiss, Poznan, Poland). The area of the wound was quantified using (1 July 2022) ImageJ software (https://imagej.nih.gov/ij/). The cell migration was expressed as the ratio of wound closure (R): R = [(A _0 h_ − A _∆ h_)/A _0 h_] where, A _0 h_ is the area of the cell-free gap measured immediately after the insert was removed, and A _∆ h_ is the area of the artificial wound measured after 12, 24 or 48 h.

### 3.8. Statistical Analysis

Unless otherwise stated, all experiments were repeated at least three times and analyzed using an independent *t* test (Statistica 13.1). The *p*-value ≤ 0.05 was considered to indicate a statistically significant difference.

## 4. Discussion

Reprogramming of glucose metabolism has been officially recognized for more than a decade as one of the most important features of cancer, characterized by the fact that under normoxia, pyruvate, instead of being transported to the mitochondrion, where it is the main fuel input for oxidative phosphorylation, enters the less energy-efficient process of lactate production [26]. In this process, LDHA is a key player responsible for the production of ATP and the regeneration of oxidized NAD, factors necessary for the processes of synthesis that enable cancer cell proliferation and invasion [27]. There are a number of studies indicating that LDHA expression is elevated in a wide variety of cancers, including colorectal cancer, breast cancer, pancreatic cancer, and oral squamous cell carcinoma, and that its level is closely correlated with tumor progression [28,29,30,31]. We, in our work using the Gene Expression database of Normal and Tumor tissues 2 (GENT2), which is a rich collection of gene expression patterns in various normal and tumor tissues compiled from public gene expression datasets, found that also in esophageal cancers LDHA expression is significantly higher, compared to normal controls, which is in line with previous reports [32]. In immunohistochemical images of esophageal cancer tissues from patients, we observed high staining intensity for LDHA but variable intensity within the same clusters of tumor cells, which may be related to their degree of differentiation. Nevertheless, these results considered altogether confirm the significant contribution of LDHA to cancer development in the esophagus.

Esophageal tumors, whether they grow from squamous or glandular epithelium, are characterized by high aggressiveness and rapid metastasis formation. Both the processes of cancer cell invasion and metastasis require the active movement of cells. In this context, the search for ways to reduce the migratory capacity of tumor cells seems necessary [33]. TNF-α, a pleiotropic cytokine produced mainly by activated macrophages, T lymphocytes, and natural killer (NK) cells, but also by tumor cells themselves, is one of the common and important components of the tumor microenvironment known to promote tumor cell migration and invasion through activation of NFκB and ERK/MAPK signaling pathways [8,34,35]. The results of our previous studies clearly indicated that TNF-α induces an increase in MMP9 expression, which has been shown to be crucial for cell migration [8,36]. The current study showed that TNF-α could also moderately modulate glucose metabolism by increasing the expression of LDH subunits in esophageal cancer cells. This is consistent with research by Vaughan et al., using breast cancer cells, which showed that TNF-α acts as a direct inducer of the Warburg effect, causing an increase in glucose uptake, lactate release, and glucose transporter 1 (GLUT1) expression [14]. In our study, a TNF-α-induced increase in lactic acid secretion was also observed, but mainly in EC7 cells in which only the LD5 isoform was active. It is known that the LDHA subunit has a higher affinity for pyruvate, so LDH isoforms rich in these peptides, especially LD5, preferentially convert pyruvate to lactic acid. In KYSE150, TNF-α stimulated an increase in the expression of A and B subunits at similar levels and, as a result, did not disrupt the LDH isoenzyme profile, which was probably the reason that the changes in lactic acid production were negligible in these cells. While it is known that TNF-α can modulate LDHA expression independently of hypoxia through hypoxia-inducible factor-1α (HIF-1α), the level of which increases in the presence of TNF-α, which we also observed in our study, the mechanism of regulation of the expression of the gene encoding the LDHB subunit by TNF-α is unknown and requires additional studies [37]. Although overall, the effect of TNF-α on LD5-dependent metabolic activity in esophageal cancer cells does not appear to be of great significance, our results strongly suggest that the presence of this cytokine enhances the Warburg effect in tumors.

A close relationship between the metabolic status of esophageal cancer cells and the pro-tumor effects of TNF-α was confirmed when cells were treated with SO. As an analog of pyruvate, SO competitively inhibits the activity of LDH isoforms that use pyruvate as a substrate and effectively disrupt glycolysis pathways. Although SO did not affect the viability of the cells used in this study, which is consistent with other studies, it definitely had an impact on their migratory potential [38,39]. It completely inhibited the promigratory effect of TNF-α, which, as our previous studies demonstrated, significantly depends on MMP9 [8,22,36]. Here we observed that SO reduced MMP9 expression in response to TNF-α as a result of inhibition of ERK1/2 signaling pathway activation. Similar results were obtained when LDHA gene expression was silenced using siRNA, although in this case, the observed effects on both cell migration and MMP9 expression were much weaker. The reason may be the limited efficiency of the siRNA method used. On the other hand, SO was demonstrated to inhibit pyruvate carboxylation or aspartate aminotransferase (AAT) activity, so the extent of its effect on glucose metabolism is not limited solely to blocking the conversion of pyruvate to lactate [40,41].

Most importantly, in this study, we found that SO, which, although it did not induce cancer cell death, significantly reduced lactic acid secretion by both KYSE150 and EC7 cells. There is evidence that cancer cells produce up to ten times more lactic acid than normal cells, and increasing concentrations of lactate in tumor tissues have been shown to correlate positively with the risk of distant metastasis [42,43,44]. LA has been found to modulate a number of cellular activities, acting as an alternative energy source, signaling molecule, and immunosuppressive agent [23,45,46,47,48]. By lowering the extracellular pH, LA stimulates the production and secretion of hyaluronic acid by cancer-associated fibroblasts (CAFs) and thus participates in the reorganization of the extracellular matrix and facilitates cell migration [49]. Here, we found that lactate can also enhance the promigratory effect of TNF-α in esophageal cancer cells by intensifying TNF-α-related MMP9 expression, and the use of SO stopped this activity. SO has so far only been used in preclinical studies and is not a Food and Drug Administration (FDA)-approved drug, but it is considered a low-toxicity compound and, therefore, relatively safe. On its own, SO is considered an ineffective anticancer drug, but there are many studies that have shown that, when combined with clinically used chemotherapeutics, it acts as a synergizing agent [38,39,50].

## 5. Conclusions

We revealed for the first time that LDHA inhibition blocks TNF-α-induced migration of esophageal cancer cells and effectively inhibits MMP9 expression in cells stimulated with TNF-α by attenuating activation of the ERK1/2-related signaling pathway. Thus, we confirmed a direct link between the metabolic status of glucose and the efficacy and specificity of the tumor cell response to the tumorigenic effects of TNF-α, one of the major pro-inflammatory cytokines accompanying cancer growth. Our results highlight the efficacy of SO in inhibiting lactic acid production by esophageal cancer cells and thus show that the use of an LDHA inhibitor can be considered in the design of adjuvant therapy for esophageal cancer patients.

## Figures and Tables

**Figure 1 ijms-23-16062-f001:**
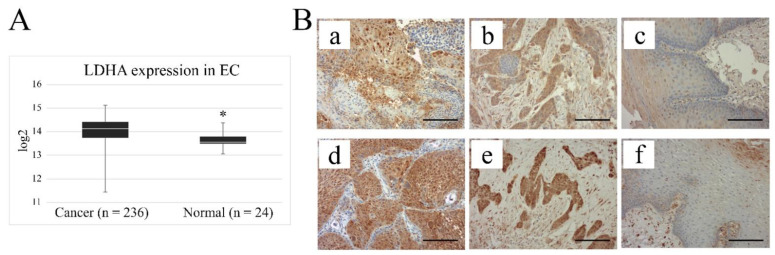
Expression levels of LDHA in cancerous and normal esophageal tissues. (**A**) Comparative analysis of LDHA gene expression in normal (n = 24) and tumor (n = 236) esophageal tissues based on data downloaded from gent2 database. Results show significantly higher levels of LDHA mRNA in tumors, compared to the control group. (**B**) Representative examples of LDHA localization in histologically normal epithelium taken from the border of resection from patients operated for esophageal cancer (**c**,**f**), and in ESCC tissues (**a**,**d**) and EAC tissues (**b**,**e**), visualized by immunohistochemical (IHC) method (magnification: ×200). Scale bars, 100 µm. * *p* < 0.005.

**Figure 2 ijms-23-16062-f002:**
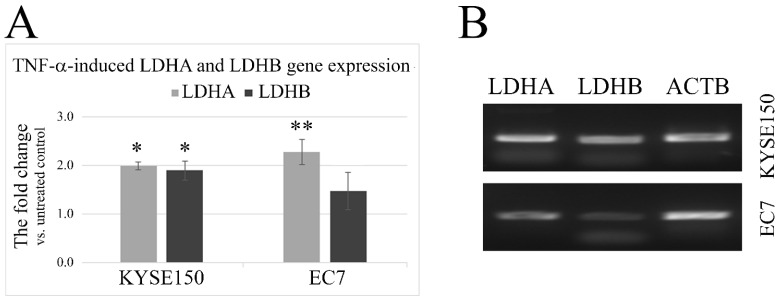
Expression levels of LDHA and LDHB in esophageal cancer cells. (**A**) RT-qPCR analysis of LDHA and LDHB expression changes in response to TNF-α (30 ng/mL) stimulation for 24 h in ESCC and EAC cells (KYSE150 and EC7, respectively). Statistically significant increase in TNF-α-dependent expression for both LDHA (FC = 1.98 ± 0.08 vs. untreated control) and LDHB (FC = 1.88 ± 0.2 vs. untreated control) was found in KYSE150 cells. In the presence of TNF-α in EC7 cells, only LDHA levels significantly increased (FC = 2.28 ± 0.26 vs. untreated control). (**B**) LDHA and LDHB mRNA expression in two esophageal cancer cell lines, KYSE150 and EC7, by regular RT-PCR. ACTB (β-actin) was used as an internal control. * *p* < 0.05, ** *p* < 0.001.

**Figure 3 ijms-23-16062-f003:**
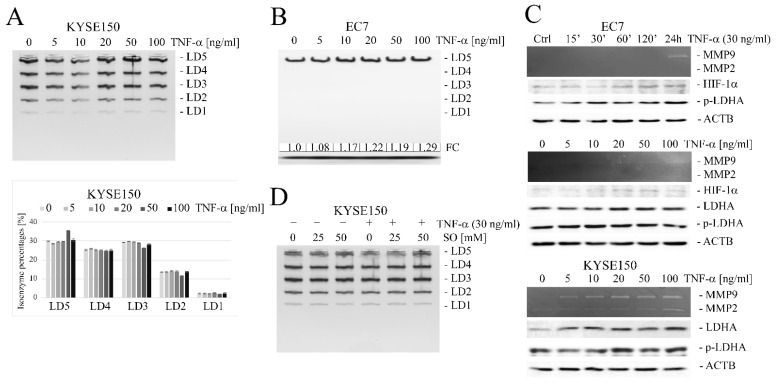
The effect of TNF-α on the composition of LDH isoforms and on the protein levels of LDHA and MMP9 in esophageal cancer cells. (**A**) Electrophoretic patterns in KYSE150 cells stimulated with TNF-α at different concentrations (upper panel) and the corresponding isoform profiles (lower panel) showing no change in the percentage of LDH isoforms. (**B**) Dose-dependent changes in electrophoretic patterns in EC7 cells under TNF-α stimulation; FC—fold change vs. untreated control calculated from differential densitometry of signals from the gel. (**C**) Changes in MMP9 activity by gelatin zymography and protein levels of LDHA, LDHB, HIF-1α, phospho-LDHA(Tyr10), LDHA by Western blotting in EC7 (upper panel) and KYSE150 (lower panel), respectively, in relation to TNF-α concentration and time. (**D**) Electrophoretic pattern in TNF-α (30 ng/mL) stimulated KYSE150 for 24 h in the presence of SO (25 and 50 mM) showing no change in the relative percentage of isoenzymes. Ctrl—control, LD1-LD5—LDH isoenzymes, ACTB—β-actin/loading control.

**Figure 4 ijms-23-16062-f004:**
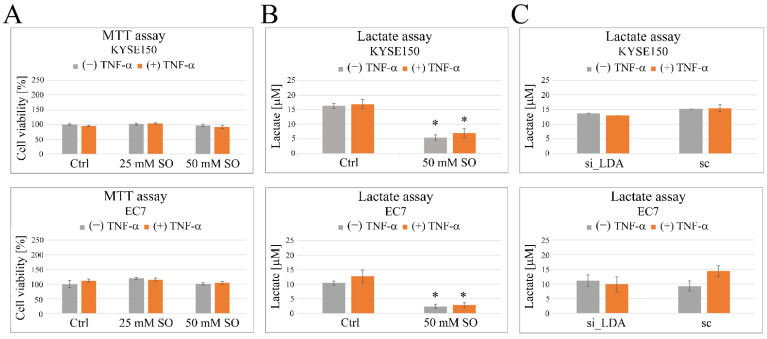
Effect of LDHA inhibition on esophageal cancer cell viability and lactic acid (LA) production. (**A**) The viability of both squamous epithelial cancer cells and cells of adenocarcinoma origin after 48 h exposure to SO at concentrations of 25 mM and 50 mM was measured using the MTT assay. Under the conditions used, SO did not significantly affect the viability of KYSE150 (upper panel) or EC7 (lower panel) cells regardless of whether they were additionally stimulated with TNF-α or not. (**B**) SO effectively inhibited LA secretion in both cell types, regardless of the presence of TNF-α, and the observed differences from the untreated control were statistically significant. (**C**) No statistically significant differences in LA production were observed in both KYSE150 (upper panel) and EC7 (lower panel) cells with siRNA-mediated silencing of LDHA gene expression. Ctrl—control, si_LDA—LDHA siRNA, sc—scrambled sequence siRNA, SO—sodium oxamate. * *p* < 0.005.

**Figure 5 ijms-23-16062-f005:**
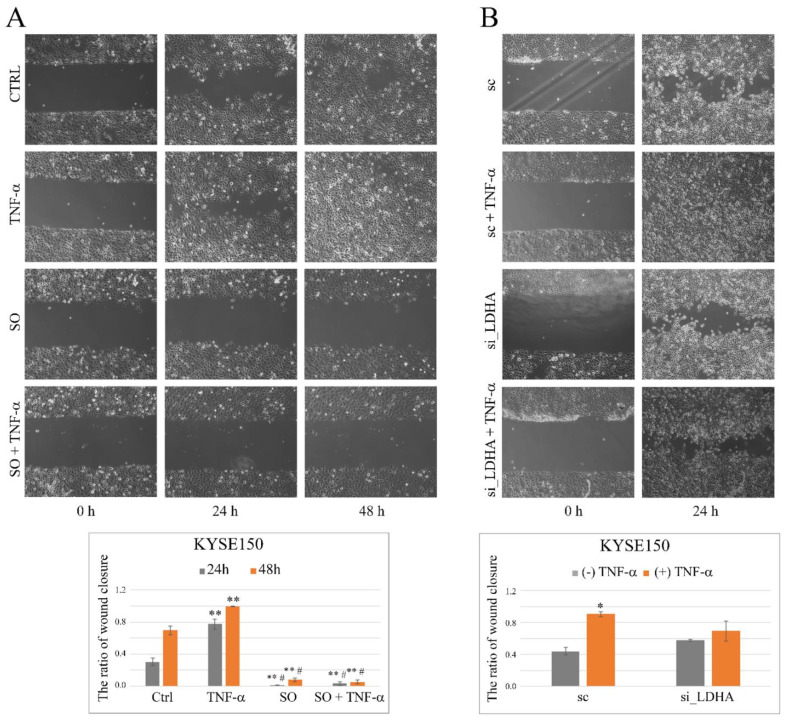
Effect of sodium oxamate and LDHA siRNA on TNF-α-induced migration of esophageal cancer cells. (**A**) Representative images from wound healing assay showing the changes in KYSE150 cell migration under SO and TNF-α applied separately and in combination after 24 h and 48 h of treatment. Quantitative analysis from the wound healing study in the form of a bar graph illustrating the ratio of wound closure in control cells and TNF-α stimulated cells for 24 h and 48 h in the presence of SO and without SO (bottom). (**B**) Representative images from the wound healing assay showing changes in KYSE150 cells with siRNA-mediated silencing of LDHA gene expression and treated with TNF-α for 24 h. Quantitative analysis from the wound healing assay in the form of a bar graph showing the ratio of wound closure after LDHA siRNA (si_LDHA) transfection or scrambled sequence (sc) and 24 h TNF-α stimulation (bottom). Ctrl—control, SO—sodium oxamate, * *p* < 0.05, ** *p* < 0.005 (statistical differences vs. controls); # *p* < 0.005 (statistical differences vs. TNF-α-treated cells).

**Figure 6 ijms-23-16062-f006:**
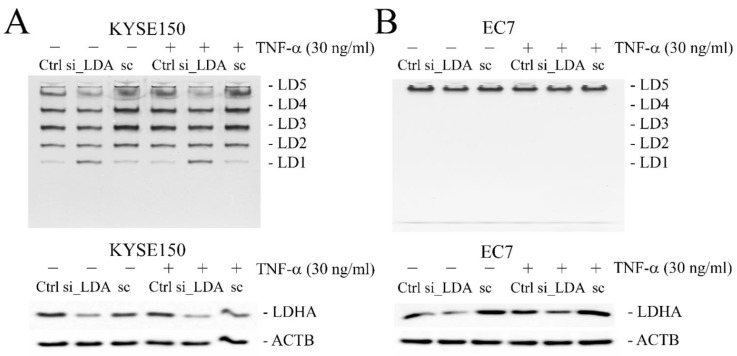
The effect of LDHA gene silencing on the composition of LDH isoforms and on the protein level of LDHA in esophageal cancer cells. (**A**) A shift in the percentage of LDH isoforms toward forms with an increased proportion of the B subunit, observed in KYSE150 cells transfected with LDHA siRNA by zymography (upper panel). Downregulation of LDHA protein in KYSE150 cells due to silencing of LDHA gene expression (SE: 38 ± 1.5%) visualized by Western blotting (bottom panel). (**B**) Reduced levels of LD5 isoform, as seen on zymogram (upper panel), and LDHA protein, as observed by Western blotting (bottom panel), as a result of silencing of LDHA gene expression (SE: 55.7 ± 13%) in EC7 cells. Ctrl—control; si_LDA—LDHA siRNA; sc—scrambled sequence siRNA; ACTB—β-actin/loading control; SE—gene silencing efficiency vs. control levels.

**Figure 7 ijms-23-16062-f007:**
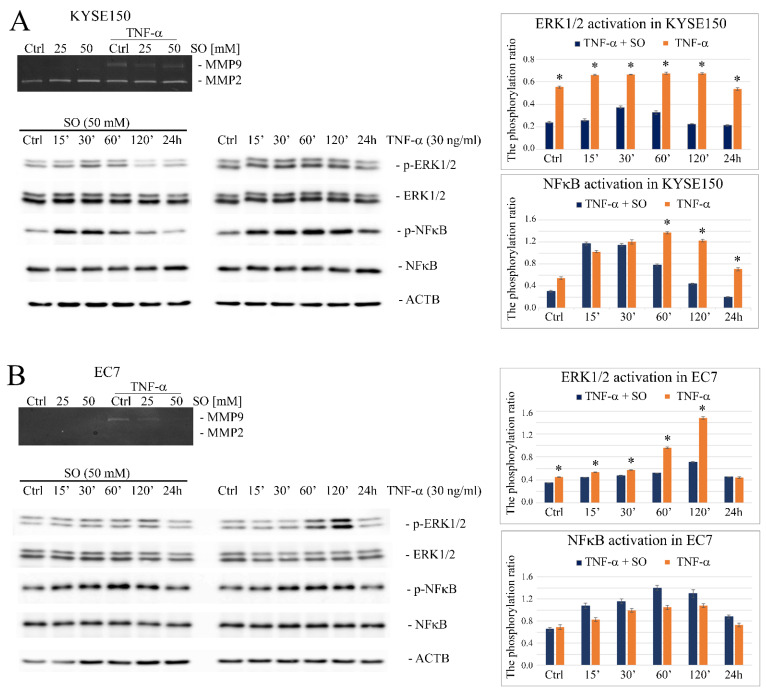
The effect of sodium oxamate on activation of TNF-α-induced MMP9-related ERK1/2 and NFκB signaling pathways in esophageal cancer cells. SO-dependent downregulation of MMP9 in response to TNF-α, seen in gelatin zymography (upper), and time corse of ERK1/2 and NFκB/p65 phosphorylation after TNF-α stimulation in (**A**) KYSE150 cells and (**B**) EC7 cells treated or not with SO (50 mM) for 24 h, visualized by Western blotting. Quantitative analysis using the differential densitometry of TNF-α-induced ERK1/2(Thr202/Tyr204) and NFκB/p65(Ser536) activation in the form of bar graphs, showing the ratio of phospho-proteins normalized to total protein in SO-treated and non-treated cells (right panels). Ctrl—control; SO—sodium oxamate; ACTB—β-actin/loading control; * *p* < 0.001.

**Figure 8 ijms-23-16062-f008:**
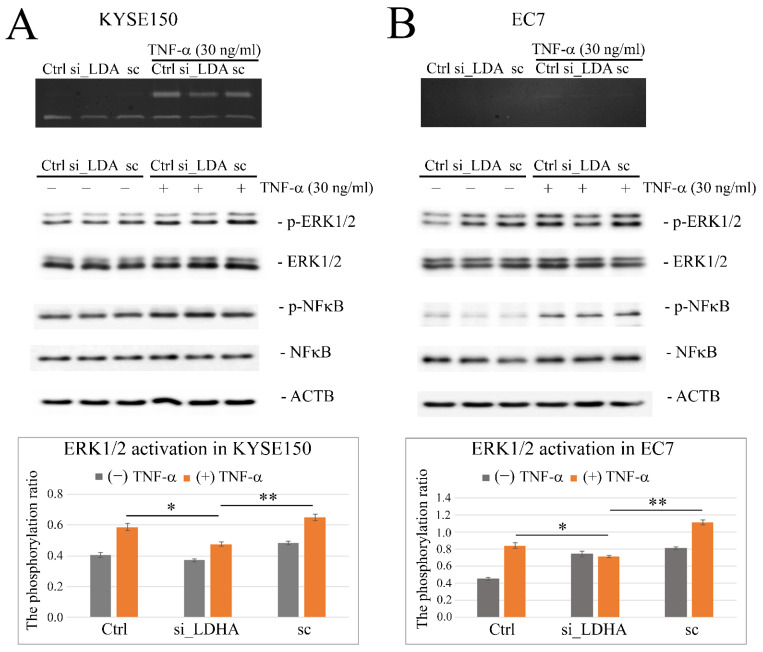
The effect of LDHA gene silencing on activation of TNF-α-induced MMP9-related ERK1/2 and NFκB signaling pathways in esophageal cancer cells. LDHA siRNA-associated downregulation of MMP9 in response to TNF-α in KYSE150 cells, seen in gelatin zymography (upper) and levels of total and phosphorylated ERK1/2 and NFκB/p65 proteins in (**A**) KYSE150 cells (**B**) and EC7 cells transfected with LDHA siRNA, subjected or not to hourly TNF-α stimulation (30 ng/mL), visualized by Western blotting. Quantitative analysis using the differential densitometry of TNF-α-induced ERK1/2 activation in the form of bar graphs, showing the ratio of phospho-ERK1/2 (Thr202/Tyr204) normalized to total ERK1/2 in LDHA siRNA-transfected and control cells (bottom panels). Ctrl—control; si_LDHA—LDHA siRNA; sc—scrambled sequence siRNA; * *p* < 0.05; ** *p* < 0.005.

**Figure 9 ijms-23-16062-f009:**
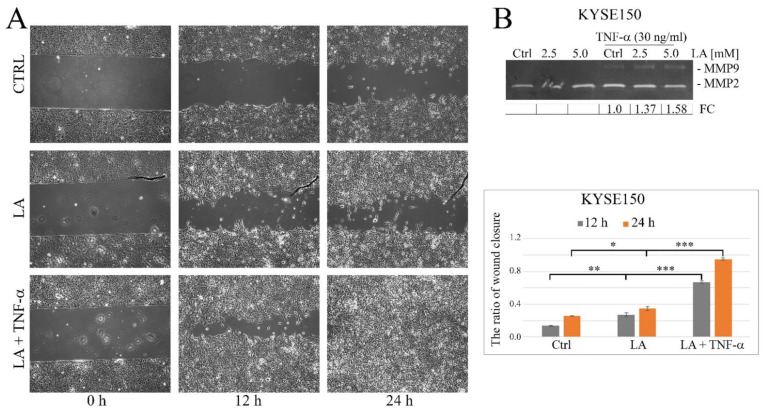
Effect of sodium lactate (LA) on TNF-α-induced MMP9 expression and cell migration in esophageal cancer cells. (**A**) Representative images from wound healing assay showing the changes in KYSE150 cell migration under LA applied separately and in combination with TNF-α after 12 h and 24 h of treatment. Quantitative analysis from the wound healing assay showing the ratio of wound closure in control cells and LA-treated cells, stimulated or not with TNF-α for 12 h and 24 h (bar graph). (**B**) LA-associated enhancement of TNF-α-induced upregulation of MMP9 in KYSE150 cells, as seen by gelatin zymography. FC—fold change calculated from differential densitometry of signals from two zymograms; Ctrl—control; LA—lactic acid; * *p* < 0.05, ** *p* < 0.005, *** *p* < 0.0005.

## Data Availability

Not applicable.

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
