# Peer review of "Effect of LDHA Inhibition on TNF-α-Induced Cell Migration in Esophageal Cancers"

_ijms, 2022, doi:10.3390/ijms232416062_

Round 1

Reviewer 1 Report

The topic of the manuscript is interesting.

I suggest the use of a plagiarism detector.

Among the 51 references, at least six are from the authors.

I propose using the complete name of proteins/genes when it appears for the first time and then the respective abbreviations. For instance, in the abstract, TNF-α, among others.

I recommend the revision of the English spelling and grammar. Some sentences are quite extensive.

For example, in the abstract:

“It is also known that TNF-α is able to induce the Warburg effect in some cells by increasing glucose uptake and enhancing the expression and activity of lactate dehydrogenase subunit A (LDHA). Therefore, the present study aims (…).”

I suggest the replacement of the word “discovered” for another. The results have demonstrated… Something like this.

I greatly encourage the authors to review the results sections. The figures and graphics are too small, with poor resolution in some cases. Moreover, some are confusing, and some information is missing (time in the x-axis in fig. 7).

The authors did not present the quantification results in Fig. 6. In B, it seems in the Zymography that the levels of L5 are similar between the different experiments.

Some results/figures did not provide statistical analysis.

A schematic illustration of the mechanisms proposed would be of great interest.

Please pay attention to ethical guidelines.

Author Response

We sincerely appreciate the thorough reviewing, the valuable comments and constructive suggestions provided by the reviewers. We have addressed all concerns of both reviewers and revised the manuscript accordingly. We have highlighted all changes within the manuscript using the "Track Changes".

Here is a point-by-point response to the reviewers’ comments and concerns.

Reviewer #1:

R: I propose using the complete name of proteins/genes when it appears for the first time and then the respective abbreviations. For instance, in the abstract, TNF-α, among others.

A: It has been added.

R: I recommend the revision of the English spelling and grammar. Some sentences are quite extensive. For example, in the abstract: “It is also known that TNF-α is able to induce the Warburg effect in some cells by increasing glucose uptake and enhancing the expression and activity of lactate dehydrogenase subunit A (LDHA). Therefore, the present study aims (…).”

A: The manuscript has undergone linguistic proofreading by a qualified native speaker.

R: I suggest the replacement of the word “discovered” for another. The results have demonstrated… Something like this.

A: It has been changed.

R: I greatly encourage the authors to review the results sections. The figures and graphics are too small, with poor resolution in some cases. Moreover, some are confusing, and some information is missing (time in the x-axis in fig. 7).

A: All figures have been revised and re-described with larger fonts. Some information has been added in figure legends for greater clarity. Since the analysis of the effect of SO on the activation of TNF-induced MMP9 signaling pathways associated with ERK1/2 and NFκB was carried out over a wide time range (from 15 minutes to 24 hours), we decided to place the values corresponding to each time interval on the x-axis of the graphs in Fig.7, which are marked with an apostrophe (') for minutes and (h) for hours.

R: The authors did not present the quantification results in Fig. 6. In B, it seems in the Zymography that the levels of L5 are similar between the different experiments.

Some results/figures did not provide statistical analysis.

A: The changes in LD5 activity observed by zymography are indeed negligible. This is an effect of both the low transfection efficiency of EC7 cells, due to the fact that they are cancer stem cells, and the very high sensitivity of the zymography method itself. Therefore, we decided to omit the densitometric analysis of the gel in this case. The efficiency of silencing protein expression using siRNA for the LDHA gene in both cell lines has been included in the figure legend.

R: A schematic illustration of the mechanisms proposed would be of great interest.

A: We thank the reviewer very much for this suggestion. However, the results from this study allow us to propose only a small part of the mechanism. To present it in more detail, which is our intention, further research is needed, which we are currently conducting.

R: Please pay attention to ethical guidelines.

A: Both the experimental work, analysis of the results and their presentation have been conducted in accordance with generally applicable ethical norms and standards.

Reviewer 2 Report

The paper of Augoff and collaborators reports the results of the study on the interrelationship between the expression and activity of lactate dehydrogenase subunit A (LDHA) and TNFα-induced pro-migratory activity in esophageal cancer cells. The work clearly demonstrates that the inhibition of LDHA by sodium oxamate (or by using siRNA-mediated gene silencing) impairs a TNFα-dependent tumor cell migration. The authors deeply investigate the role of sodium oxamate (SO), and demonstrate that it inhibits MMP9 expression by affecting TNFα-induced ERK1/2 phosphorylation. Moreover, the competitive inhibition of the activity of LDH isoforms, significantly reduces lactic acid secretion, in both esophageal cancer cell lines used in this study. The authors also conclude that an LDHA inhibitor (such as SO used in this study) able to disrupt glycolysis pathways, could be considered as an adjuvant for esophageal cancer therapy.

To my opinion the conclusions are well supported by the experimental data and the work is solid. The article is well-written and easy to read. Consequently, I would suggest publication in IJMS.

 Only a minor point:

At line 23 the authors write:  sodium oxamate (SO, Aminooxoacetic acid, Oxamic acid)”. Aminooxoacetic acid and oxamic acid are not synonymous of sodium oxamate, because they are the names of the acidic form and not of the salt. If the authors want to report all the synonymous, they should use “Aminooxoacetic acid sodium salt, Oxamic acid sodium salt”.

Author Response

We sincerely appreciate the thorough reviewing, the valuable comments and constructive suggestions provided by the reviewers. We have addressed all concerns of both reviewers and revised the manuscript accordingly. We have highlighted all changes within the manuscript using the "Track Changes".

Reviewer #2:

R: At line 23 the authors write: “sodium oxamate (SO, Aminooxoacetic acid, Oxamic acid)”. Aminooxoacetic acid and oxamic acid are not synonymous of sodium oxamate, because they are the names of the acidic form and not of the salt. If the authors want to report all the synonymous, they should use “Aminooxoacetic acid sodium salt, Oxamic acid sodium salt”.

A: Thank you very much for this valuable substantive comment. Sodium oxamate synonyms have been corrected in the text.

Round 2

Reviewer 1 Report

Dear authors,

I would like to recommend to check for some missing information regarding statistical analysis of the results. In some cases, the info does not appears included in the graphs. Moreover, the figures the graphs' caption continues to be a little bit difficult to read.

Author Response

Reviewer #1: I would like to recommend to check for some missing information regarding statistical analysis of the results. In some cases, the info does not appears included in the graphs. Moreover, the figures the graphs' caption continues to be a little bit difficult to read.

Authors: we sincerely appreciate the reviewer's time spent in carefully reviewing our manuscript. References to statistical analyses that we inadvertently omitted from the figures have been completed. To improve the readability of the drawings, we changed the resolution of the images to 600 DPI and additionally enlarged the font where possible and redescribed the numbers on the Y axis.

Thank you again for all your comments and suggestions. We are convinced that they have contributed to improving the quality of this work.

Sincerely,

Katarzyna Augoff